# H+: An Efficient Similarity-Aware Aggregation for Byzantine Resilient Federated Learning

## Abstract

Federated Learning (FL) enables decentralized model training without sharing raw data. However, it remains vulnerable to Byzantine attacks, which can compromise the aggregation of locally updated parameters at the central server. Similarity-aware aggregation has emerged as an effective strategy to mitigate such attacks by identifying and filtering out malicious clients based on similarity between client model parameters and those derived from clean data, i.e., data that is uncorrupted and trustworthy. However, existing methods adopt this strategy only in FL systems with clean data, making them inapplicable to settings where such data is unavailable. In this paper, we propose H+, a novel similarity-aware aggregation approach that not only outperforms existing methods in scenarios with clean data, but also extends applicability to FL systems without any clean data. Specifically, H+ randomly selects $r$-dimensional segments from the $p$-dimensional parameter vectors uploaded to the server and applies a similarity check function $H$ to compare each segment against a reference vector, preserving the most similar client vectors for aggregation. The reference vector is derived either from existing robust algorithms when clean data is unavailable or directly from clean data. Repeating this process $K$ times enables effective identification of honest clients. Moreover, H+ maintains low computational complexity, with an analytical time complexity of $\mathcal{O}(KMr)$, where $M$ is the number of clients and $Kr \ll p$. Comprehensive experiments validate H+ as a state-of-the-art (SOTA) method, demonstrating substantial robustness improvements over existing approaches under varying Byzantine attack ratios and multiple types of traditional Byzantine attacks, across all evaluated scenarios and benchmark datasets.

## 1 Introduction

Federated Learning (FL) has emerged as a distributed paradigm to address challenges related to large-scale data and privacy. It enables edge clients to collaboratively train a global model without sharing raw data (Zuo et al., 2025; Konečnỳ et al., 2016; Wang et al., 2019). Within the FL framework, a central server coordinates with clients by exchanging model parameters or gradient vectors instead of raw data, thereby advancing the learning process (Guo et al., 2023; Xiao & Ji, 2023). This privacy-preserving mechanism, combined with the growing capabilities of edge computing, has made FL increasingly appealing in modern machine learning scenarios (Dorfman et al., 2023).

While the distributed nature of FL brings notable advantages in efficiency and privacy, it also introduces robustness challenges that have drawn increasing attention due to the participation of numerous clients (Yang et al., 2020; Pang et al., 2023; Vempaty et al., 2013). The vectors uploaded to the central server may include irrelevant or erroneous information, arising from heterogeneous data distributions, client-device inconsistencies, or even malicious behavior (So et al., 2020). Clients that intentionally submit false or harmful information are referred to as Byzantine clients, while the rest are considered honest participants (Chen et al., 2017). During training, Byzantine clients can adaptively generate and coordinate deceptive model updates, severely degrading the performance of the global model (Cao & Lai, 2019). Therefore, enhancing the robustness of FL systems against Byzantine attacks has become a pressing security concern in distributed learning frameworks (Kairouz et al., 2021).

A key metric for evaluating robustness against Byzantine attacks in FL is the maximum Byzantine client ratio that an aggregation method can tolerate while still achieving satisfactory model performance, such as high test accuracy (Xie et al., 2018; Blanchard et al., 2017). In conventional FL settings without assumed clean data, most existing defenses mitigate malicious clients by leveraging statistical or geometric properties of their updates (Pillutla et al., 2022; Karimireddy et al., 2021). These methods typically require that the majority of clients be honest, limiting the tolerable Byzantine ratio to under 0.5 (Luan et al., 2024). Once this threshold is exceeded, purely algorithmic defenses based on parameter statistics often fail to provide reliable robustness guarantees. To relax this fundamental limitation, recent approaches introduce the notion of clean data, which may reside at the server or at subset of trusted clients (Regatti et al., 2020). Leveraging clean data enables the system to evaluate the consistency of received updates and distinguish between benign and adversarial behavior (Xie et al., 2020b). Among these techniques, similarity-aware aggregation has shown promise by identifying and downweighting client updates that deviate from patterns observed in clean data. This class of methods enhances robustness even under high Byzantine ratios, provided that reliable reference data is accessible. Existing similarity-aware aggregation methods, such as Xie et al. (2020b), which utilize cosine similarity to filter honest clients more efficiently than non-similarity-aware counterparts, operating with computational complexity linear in the model parameter dimension $p$, but may fail on large $p$ due to the curse of dimensionality in similarity measurement (Hastie et al., 2009).

Additionally, despite their effectiveness, such similarity-based strategies have not been widely adopted in FL systems where clean data is unavailable. Some prior works attempt to detect and exclude Byzantine clients through unsupervised techniques or client clustering (Blanchard et al., 2017), but these methods often fail to achieve acceptable performance across various attack types and under high Byzantine ratios.

The above limitations highlight the necessity for a unified robust aggregation framework that not only overcomes the challenges faced by existing similarity-aware methods in clean data settings but also extends their applicability to scenarios where clean data is unavailable. In this paper, we propose a novel similarity-aware aggregation method tailored for FL settings with or without access to clean data. To reduce computational overhead, each uploaded $p$-dimensional model update is randomly partitioned into multiple $r$-dimensional segments. These segments are then evaluated using a newly designed similarity metric, denoted as the $H$ function, which measures their alignment with a reference vector. The construction of the reference vector is adaptive to the availability of clean data: when clean data is available, it is directly derived from the corresponding segments of trusted sources; otherwise, it is obtained through existing robust aggregation techniques. By performing similarity evaluations across multiple segments, the method identifies a stable intersection set of clients whose updates consistently resemble the reference. Only these clients deemed potentially honest are selected for final aggregation, enhancing robustness against Byzantine behaviors while maintaining computational efficiency. The main contributions of our proposed H+ method are summarized as follows:

- We propose H+, a novel Byzantine-resilient aggregation method that leverages similarity awareness and is applicable to FL system both with and without access to clean data. H+ generalizes the core idea of identifying Byzantine clients based on similarity, from previously relying on clean data to scenarios where no clean data is available. In clean-data settings, H+ operates as a standalone aggregation algorithm. In the absence of clean data, H+ serves as a lightweight plug-in module that complements existing robust aggregation methods by utilizing their outputs to construct reference vectors for similarity evaluation.

- From a computational perspective, H+ achieves a complexity of $\mathcal{O}(KMr)$, where $Kr \ll p$, significantly reducing the overhead compared to existing similarity-aware aggregation methods designed for settings with clean data. Moreover, in scenarios without clean data, H+ introduces only minimal additional computation, as it reuses outputs from existing robust algorithms. This lightweight design ensures scalability and makes H+ particularly well-suited for large-scale FL models.

- Extensive experiments on benchmark datasets with heterogeneous data distributions show that H+ consistently achieves state-of-the-art (SOTA) performance in terms of test accuracy across a wide range of Byzantine attack types and attack ratios, under both clean-data

and no-clean-data settings. These results demonstrate the superior robustness of H+ over existing aggregation methods in diverse and adversarial federated learning environments.

## 2 RELATED WORK

### 2.1 ROBUST AGGREGATION METHODS WITHOUT CLEAN DATA

In this area, existing methods generally fall into two categories: selection-based approach represented by Krum that aims to identify and exclude Byzantine clients, and aggregation-based approaches that mitigate their influence without explicit client selection, including point-wise median, geometric median (GM), and some others. Detailed description of them are as follows: **Krum (the selection-based approach):** Blanchard et al. (2017) proposes selecting the uploaded vector with the shortest Euclidean distance to all others for global updates; it also introduces Multi Krum, which applies Krum iteratively to counter attacks.

In the context of aggregation-based approaches, existing methods include **Median:** The earliest work using median to resist Byzantine attacks is Xie et al. (2018), which computes the point-wise median of uploaded vectors as the aggregation vector for global model updates. Building on this, Yin et al. (2018) selectively aggregates via point-wise trimmed mean or median to enhance Byzantine robustness. **GM:** Robust Federated Aggregation (RFA) (Pillutla et al., 2022), Byzantine-resilient distributed Stochastic Average Gradient Algorithm (Byrd-SAGA) (Wu et al., 2020), and Byzantine-RObust Aggregation with gradient Difference Compression And STochastic variance reduction (BROADCAST) (Zhu & Ling, 2023) all adopt GM to boost FL robustness. RFA uses the tail-average of local parameters as uploaded vectors; Byrd-SAGA leverages the SAGA method (Defazio et al., 2014) for global updates; BROADCAST extends Byrd-SAGA by incorporating quantization. **Other methods:** Robust Stochastic Aggregation (RSA) (Li et al., 2019) uses $l$-norm to penalize differences between local and global parameters, isolating Byzantine clients. Maximum Correntropy Aggregation (MCA) (Luan et al., 2024) aggregates vectors via maximum correntropy. Centered Clipping (CClip) (Karimireddy et al., 2021) clips the magnitude of uploaded vectors using previously aggregated vectors.

### 2.2 ROBUST AGGREGATION METHODS WITH CLEAN DATA

**Non-similarity-aware method:** Zeno (Xie et al., 2019) formulates a stochastic descent score, which calculated from the global model and clean data, to filter honest vectors, while Zeno+ (Xie et al., 2020b) extends Zeno to asynchronous settings. Cao & Lai (2019) uses a vector derived from clean data to filter honest uploads via a modulus-bounded approach. By contrast, ByGARS (Regatti et al., 2020) leverages a vector generated by clean data to adjust reputation scores, differing slightly from Cao & Lai (2019). **Similarity-aware method:** FLTrust (Cao et al., 2021) utilizes the cosine similarity between a reference vector (calculated from clean data) and the uploaded vectors to aggregate these uploaded vectors via a weighted average. And Zeno++ (Xie et al., 2020b) further refine this method by improving stochastic descent score generation with cosine similarity for asynchronous settings, outperforming non-similarity-aware methods in efficiently and effectively boosting FL performance and robustness. However, cosine similarity is computationally expensive and may still fail to detect honest clients for large $p$, as it tends to zero in high dimensions.

## 3 PROBLEM SETUP

### 3.1 FL OPTIMIZATION PROBLEM

Consider an FL system with one central server and $M$ clients, which form the set $\mathcal{M} \triangleq \{1, 2, 3, \cdots, M\}$. For any participating client, say the $m$th client, it has a local dataset $\mathcal{S}_m$ containing $S_m$ elements. The $i$th element of $\mathcal{S}_m$ is a ground-truth sample $s_{m,i} = \{x_{m,i}, y_{m,i}\}$. Here, $x_{m,i} \in \mathbb{R}^{in}$ represents the input vector, and $y_{m,i} \in \mathbb{R}^{out}$ denotes the output vector. Using the datasets $\mathcal{S}_m$ for $m = 1, 2, 3, \cdots, M$, the learning task is to train a $p$-dimensional model parameter $w \in \mathbb{R}^p$ to minimize the global loss function, denoted as $F(w)$. Specifically, we aim to solve the following optimization problem:

$$\min_{w \in \mathbb{R}^p} F(w) \tag{1}$$

## 3.2 FL without Clean Data

For FL without clean data, the central server does not have any data, and the entire FL training optimization process relies on clients' private datasets. Hence, the global loss function $F(w)$ in (1) can be defined as

$$F(w) \triangleq \frac{1}{\sum_{m \in \mathcal{M}} S_m} \sum_{m \in \mathcal{M}} \sum_{s_{m,i} \in \mathcal{S}_m} f(w, s_{m,i}) \tag{2}$$

where $f(w, s_{m,i})$ denotes the loss function to evaluate the error for approximating $y_{m,i}$ given the input $x_{m,i}$. For convenience, we define the local loss function of the $m$th client as

$$F_m(w) \triangleq \frac{1}{S_m} \sum_{s_{m,i} \in \mathcal{S}_m} f(w, s_{m,i}) \tag{3}$$

and the weight coefficient of the $m$th client as $\alpha_m = S_m / (\sum_{m' \in \mathcal{M}} S_{m'}), m \in \mathcal{M}$. The global loss function $F(w)$ is then rewritten as

$$F(w) = \sum_{m \in \mathcal{M}} \alpha_m F_m(w) \tag{4}$$

## 3.3 FL with Clean Data

**The central server has clean data:** Consider that the central server possesses some clean data (to enhance training performance and improve robustness), forming a dataset $\mathcal{S}_0$ with $S_0$ elements. Similarly, we define the sever loss function of the central server as

$$F_0(w) \triangleq \frac{1}{S_0} \sum_{s_{0,i} \in \mathcal{S}_0} f(w, s_{0,i}) \tag{5}$$

and the weight coefficient for the central server and the $M$ clients as $\alpha'_m = S_m / (\sum_{m' \in \mathcal{M}^{\dagger}} S_{m'}), m \in \mathcal{M}^{\dagger}, \mathcal{M}^{\dagger} = \{0\} \cup \mathcal{M}$. The global loss function $F(w)$ in (1) is then rewritten as

$$F(w) = \sum_{m \in \mathcal{M}^{\dagger}} \alpha'_m F_m(w) \tag{6}$$

**The central server is aware that some clients possess clean data (a subset of honest clients is known):** Consider that the central server knows a subset of honest clients (even just one); in this case, the global loss function $F(w)$ is the same as in (4), written as follow,

$$F(w) = \sum_{m \in \mathcal{M}} \alpha_m F_m(w) \tag{7}$$

## 3.4 Byzantine Attacks

Based on the above FL frameworks, assume there are $B$ Byzantine clients among the $M$ total clients, forming the set $\mathcal{B}$. Any Byzantine client can send an arbitrary vector $\star \in \mathbb{R}^p$ to the central server. Let $g_m^t$ denote the actual vector uploaded by the $m$th client to the central server during the FL training process, then we have

$$g_m^t = \star, m \in \mathcal{B} \tag{8}$$

For ease of representing the ratio of Byzantine clients, we denote the intensity level of the Byzantine attacks as $\bar{C}$, defined by the weight coefficient of Byzantine clients as

$$\bar{C} \triangleq \begin{cases} \sum_{m \in \mathcal{B}} \alpha'_m, & \text{where the central server has clean data} \\ \sum_{m \in \mathcal{B}} \alpha_m, & \text{other cases} \end{cases} \tag{9}$$

# 4 METHODOLOGY

In this section, we first introduce the similarity check function $H$, which forms the basis of our robust method. We then explain the application of the similarity check function $H$ and our method H+ to the two FL frameworks described above.

## 4.1 SIMILARITY CHECK FUNCTION

To distinguish Byzantine attacks, we introduce a similarity check function $H$. For $\forall X, Y \in \mathbb{R}^p$, the function $H(X, Y)$ is defined as

$$H(X, Y) \triangleq \frac{1}{p} \sum_{i=1}^{p} \frac{|x_i|}{|y_i - x_i| + |x_i|} \tag{10}$$

where $X = (x_1, x_2, \cdots, x_p)^T$ and $Y = (y_1, y_2, \cdots, y_p)^T$. From the above definition of the similarity check function $H$, we can easily see that $0 \leq H \leq 1$: the closer $H$ is to 1, the greater the similarity between $X$ and $Y$. However, when $p$ is large, the cost and complexity of calculating $H$ are very high. Thus, direct application is not conducive to training current large models. **Repeated slicing of the X and Y vectors for dimension reduction not only drastically reduces computational overhead but also mitigates the curse of dimensionality in similarity measurement.** Here we design H+ method based on the similarity check function $H$ for the two FL frameworks, which are described in detail as follows.

## 4.2 H+ ON FL WITHOUT CLEAN DATA

For FL without clean data, to defend against Byzantine attacks with $\bar{C} < 0.5$, we design the H+ method, whose procedure is shown as follows:

**Local Training:** In the $t$th iteration, after receiving the global model parameter $w^t$ broadcast by the central server, all honest clients $m \in \mathcal{M} \setminus \mathcal{B}$ select a subdataset $\xi_m^t$ from their own dataset $\mathcal{S}_m$ to calculate their local training gradients $\nabla F(w^t, \xi_m^t)$. Meanwhile, all Byzantine clients $m \in \mathcal{B}$ may send arbitrary vectors or other malicious vectors based on their datasets, the global model parameter $w^t$, and other clients' local training gradients. Let $g_m^t$ denote the vector (either the local training gradient or the malicious vector) uploaded to the central server by client $m$, then we have

$$g_m^t = \begin{cases} \nabla F(w^t, \xi_m^t), & m \in \mathcal{M} \setminus \mathcal{B} \\ \star, & m \in \mathcal{B} \end{cases} \tag{11}$$

**Aggregation and Broadcasting:** In the $t$th iteration, upon receiving all vectors $g_m^t$ from clients, the central server aggregates these vectors using existing aggregation algorithms (e.g., GM or MCA). We abbreviate such aggregation algorithms as $\mathrm{AGG}(\cdot)$, and the the reference vector $g^t$ can be calculated by

$$g^t = \mathrm{AGG}(\alpha_1, \alpha_2, \cdots, \alpha_M; w_1^t, w_2^t, \cdots, w_M^t) \tag{12}$$

To enhance the robustness of these existing aggregation algorithms, we calculate the similarity check function $H$ between all uploaded vectors and $g^t$, respectively. However, for large models, a direct use of $H$ function on reference and uploaded vectors incurs a computational complexity $\mathcal{O}(pM)$, not to mention such operations has to be performed in every training round. To mitigate this overhead, we randomly select $r$-dimensional segments from the reference and uploaded vectors to compute the similarity check function $H$, denotes as $\{g^t\}_r$ and $\{g_m^t\}_r$. Additionally, to quickly filter outliers and occasional useless vectors in environments with heterogeneous data, we introduce a penalty term $\max\{\mathrm{norm}_m, \tau/\mathrm{norm}_m\}$, where $\mathrm{norm}_m$ denotes the modulus of $\{g_m^t\}_r$ and $\tau$ is a tunable hyperparameter. Based on the above discussion, the final anomaly score is defined as

$$\mathrm{score}_m = H(\{g^t\}_r, \{g_m^t\}_r) - \rho \cdot \max\{\mathrm{norm}_m, \frac{\tau}{\mathrm{norm}_m}\} \tag{13}$$

where $\rho$ is a tunable hyperparameter.

The above operation will be repeated $K$ times, and for the $k$th operation, we select the $N$ uploaded vectors with highest scores to form the client index set $\mathcal{I}_k^t$. Finally, we take the intersection of these $K$ sets as $\mathcal{I}^t$, as follows:

$$\mathcal{I}^t = \mathcal{I}_1^t \cap \mathcal{I}_2^t \cap \mathcal{I}_3^t \cap \cdots \cap \mathcal{I}_K^t \tag{14}$$

After that, using learning rate $\eta^t$, the global model parameter $w^{t+1}$ can be updated by

$$w^{t+1} = w^t - \eta^t \sum_{m \in \mathcal{I}^t} \frac{\alpha_m}{\sum_{m' \in \mathcal{I}^t} \alpha_{m'}} \cdot g_m^t \tag{15}$$

Then, the central server broadcasts the global model parameter $w^{t+1}$ to all clients in preparation for the calculation in the $t+1$th iteration. The detailed algorithm workflow is shown in Algorithm 1.

### 4.3 H+ ON FL WITH CLEAN DATA

For the FL with clean data, to defend against Byzantine attacks with $\bar{C} \geq 0.5$, we enhance the application of the similarity check function $H$ in this framework, and its procedure is shown as follows.

**Local Training:** In the $t$th iteration, all clients do the same as in the classic FL framework, and we have

$$g_m^t = \begin{cases} \nabla F(w^t, \xi_m^t), & m \in \mathcal{M} \setminus \mathcal{B} \\ \star, & m \in \mathcal{B} \end{cases} \tag{16}$$

**Aggregation and Broadcasting:** In the $t$th iteration, if the central server has clean data, it generates the server gradient vector $\nabla F_0(w^t, \xi_0^t)$ by training on the subdataset $\xi_0^t$ from dataset $\mathcal{S}_0$. The reference vector $g^t$ in the two cases (where the central server has clean data and where a subset of honest clients, denotes as $\mathcal{T}$, is known) can then be calculated by

$$g^t = \begin{cases} \nabla F_0(w^t, \xi_0^t), & \text{central server has clean data} \\ \displaystyle\sum_{m \in \mathcal{T}} \frac{\alpha_m}{\sum_{m' \in \mathcal{T}} \alpha_{m'}} \cdot g_m^t, & \mathcal{T} \text{ is known} \end{cases} \tag{17}$$

After obtaining the reference vector $g^t$, the central server performs the same operations as in the FL without clean data to form the sets $\{\mathcal{I}_k^t\}$ and $\mathcal{I}^t$. Subsequently, the global model parameter $w^{t+1}$ can be updated by

$$w^{t+1} = w^t - \eta^t \sum_{m \in \mathcal{I}^t} \frac{\alpha_m'}{\sum_{m' \in \mathcal{I}^t} \alpha_{m'}'} \cdot g_m^t \tag{18}$$

with the central server has clean data or

$$w^{t+1} = w^t - \eta^t \sum_{m \in \mathcal{I}^t} \frac{\alpha_m}{\sum_{m' \in \mathcal{I}^t} \alpha_{m'}} \cdot g_m^t \tag{19}$$

when the clean data is on some participating clients.

Upon completing iteration $t$, the central server broadcasts the global model parameter $w^{t+1}$ to all clients in preparation for the calculation in the $t+1$th iteration. The detailed algorithm workflow is shown in Algorithm 2.

### 4.4 TIME COMPLEXITY OF H+

From Algorithm 1, the overall time complexity of the complete algorithm is $\mathcal{O}(\text{existing methods}) + \mathcal{O}(KMr) + \mathcal{O}(M \log M)$ (e.g., $\mathcal{O}(\text{Median}) = \mathcal{O}(pM \log M)$, $\mathcal{O}(\text{Krum}) = \mathcal{O}(pM^2)$, and $\mathcal{O}(\text{GM}) = \mathcal{O}(pM \log^3(M\bar{C}^{-1}))$) (Cohen et al., 2016)). As shown in Algorithm 2, the time complexity of the H+ method when used independently is $\mathcal{O}(KMr) + \mathcal{O}(M \log M)$. Consequently, its computational cost can be expressed as $\mathcal{O}(KMr) + \mathcal{O}(M \log M)$. Since $Kr \gg \log M$ in most practical scenarios, the overall complexity is approximated by $\mathcal{O}(KMr)$, which is significantly lower than $\mathcal{O}(Mp)$, confirming the efficiency of the H+ method.

Table 1: The maximum test accuracy (%) for the H+ method and baselines without clean data. The best results are in **bold**, and improvements brought by H+ over the original robust methods are underlined.

| β | | 0.6 | | | | | | 0.2 | | | | | |
|---|---|---|---|---|---|---|---|---|---|---|---|---|---|
| Attack Name | Dataset | Tiny-ImageNet | | CIFAR-100 | | CIFAR-10 | | Tiny-ImageNet | | CIFAR-100 | | CIFAR-10 | |
| | $\bar{C}$ | 0.2 | 0.4 | 0.2 | 0.4 | 0.2 | 0.4 | 0.2 | 0.4 | 0.2 | 0.4 | 0.2 | 0.4 |
| Gaussian Attack | H+Median | 54.67 | 53.23 | **55.14** | **54.52** | 68.02 | **67.79** | 53.03 | **51.40** | 54.22 | **53.29** | **67.20** | 63.15 |
| | Median | 47.16 | 46.36 | 49.00 | 48.55 | **68.34** | 66.93 | 21.60 | 23.41 | 22.83 | 23.95 | 59.52 | 58.51 |
| | H+Krum | **54.81** | **53.45** | 54.97 | 54.46 | 67.80 | 67.36 | **54.16** | 51.37 | **54.44** | 53.05 | 66.96 | **66.09** |
| | Krum | 32.16 | 32.20 | 30.09 | 29.98 | 49.88 | 51.71 | 26.10 | 25.85 | 22.18 | 22.27 | 56.00 | 52.02 |
| Sign-flip Attack | H+GM | 54.30 | 53.05 | 54.75 | 53.72 | 67.99 | 67.34 | 52.65 | 51.31 | 53.71 | 52.65 | 66.02 | 64.87 |
| | GM | 42.76 | 0.33 | 35.34 | 3.29 | 49.47 | 33.24 | 29.64 | 0.06 | 23.04 | 3.18 | 36.63 | 26.70 |
| | H+MCA | 54.20 | **53.39** | 54.85 | 53.81 | 67.47 | **67.88** | **52.76** | 51.34 | 53.78 | 53.16 | 66.97 | 65.51 |
| | MCA | 0.50 | 0.50 | 1.00 | 1.00 | 10.00 | 10.00 | 0.51 | 0.50 | 1.00 | 1.00 | 10.00 | 10.00 |
| | H+CClip | **54.42** | **53.39** | **54.98** | **54.41** | **68.70** | 65.86 | 52.58 | **52.54** | **54.35** | **53.72** | **68.58** | **66.12** |
| | CClip | 36.16 | 0.43 | 22.25 | 1.17 | 11.45 | 11.62 | 13.92 | 0.41 | 2.52 | 1.09 | 10.75 | 12.35 |
| LIE Attack | H+Median | **54.61** | **53.95** | 54.66 | **54.44** | 68.09 | 67.36 | **53.86** | **52.45** | **54.83** | 53.01 | **65.74** | **64.61** |
| | Median | 46.71 | 46.76 | 48.76 | 48.95 | 66.80 | 65.45 | 22.75 | 22.21 | 25.27 | 28.74 | 62.24 | 63.39 |
| | H+CClip | 54.28 | 53.63 | **54.91** | 53.94 | **68.25** | 66.57 | 53.71 | 51.81 | 54.78 | **53.16** | 65.55 | 63.78 |
| | CClip | 45.51 | 40.96 | 45.06 | 40.99 | 28.65 | 26.89 | 41.98 | 31.79 | 40.21 | 32.50 | 20.82 | 18.24 |
| FoE Attack | H+Krum | **54.65** | **54.56** | **54.96** | **54.51** | 68.28 | **68.81** | 53.37 | **51.48** | **54.51** | **53.33** | 68.27 | 68.27 |
| | Krum | 0.33 | 0.34 | 16.81 | 8.19 | 37.11 | 12.09 | 0.35 | 0.36 | 16.38 | 5.74 | 28.73 | 11.32 |
| | H+GM | 54.04 | 53.77 | 54.78 | 54.00 | 67.66 | 67.48 | 53.07 | 49.48 | 54.16 | 14.39 | 67.94 | 60.03 |
| | GM | 42.58 | 0.34 | 35.37 | 0.74 | 12.98 | 12.54 | 29.78 | 0.35 | 1.67 | 0.70 | 15.23 | 12.22 |
| | H+MCA | 53.91 | 54.02 | 54.76 | 54.14 | 67.85 | 68.03 | **53.43** | 49.79 | 54.26 | 8.73 | 68.00 | 60.66 |
| | MCA | 0.50 | 0.50 | 1.00 | 1.00 | 10.00 | 10.00 | 0.51 | 0.50 | 1.00 | 1.00 | 10.00 | 10.00 |

## 5 EXPERIMENTS

### 5.1 IMPLEMENTATION DETAILS

**Datasets, models and hyperparameters:** We conduct experiments on Tiny-ImageNet, CIFAR-100, and CIFAR-10 datasets, utilizing the MobileNetV3 (Howard et al., 2019), VGG16 (Simonyan & Zisserman, 2014), and ResNet18 (He et al., 2016) models. For the non-IID settings, we adopt the Dirichlet ($\beta$) distribution, where the label distribution on each device follows a Dirichlet distribution and the concentration parameter $\beta$ takes values 0.6 and 0.2. And all models use the default pre-training parameters. We set $M = 50$ and fix the batch size at 32 across all experiments. The number of iterations is configured as 100 for these three datasets. More detailed are provide in Appendix D.

**Byzantine attacks:** The ratio of Byzantine attacks, $\bar{C}$, is set to 0.2, 0.4, 0.5, 0.6, 0.7, 0.8 and 0.9. We select four types of Byzantine attacks (Gaussian attack, Sign-flip attack, LIE attack (Baruch et al., 2019), and FoE attack (Xie et al., 2020a)) to verify the robustness of H+ method and baselines. Additionally, we design a specific attack (referred to as "our attack") to further validate the conclusions drawn from the ablation study. More details about these attacks are provided in Appendix D.

**Baselines:** The performance of eight methods (Our method H+, Median, Krum (Blanchard et al., 2017), GM, MCA (Luan et al., 2024), CClip (Karimireddy et al., 2021), FLTrust (Cao et al., 2021), and Zeno++ (Xie et al., 2020b)) is compared. Among these, Median, Krum, GM, MCA, and CClip utilize coordinate-wise median, Krum, geometric median, maximum correntropy aggregation, and centered clipping, respectively, to update the global model parameters over the uploaded vectors on FL without clean data. FLTrust and Zeno++ utilize the clean data on the central server. Note that Cao & Lai (2019) and ByGARS are excluded from comparison due to the lack of open-source code and their relative obsolescence. Among Zeno, Zeno+ and Zeno++, Zeno++ is evaluated as it is the latest improved version. Our H+ method is evaluated under both frameworks with and without clean data, denoted as H+(X), where X specifies the algorithm to generate the reference vector.

### 5.2 COMPARISON WITH BASELINES

In this section, we evaluate our H+ method and baselines on the Tiny-ImageNet, CIFAR-100, and CIFAR-10 datasets. Table 1 and Table 2 show that H+ improves upon existing robust methods and

Table 2: The maximum test accuracy (%) for the H+ method and Zeno++ with clean data on $\beta = 0.6$. The best results are in **bold**.

| Attack Name | Datasets | Tiny-ImageNet | | | | | CIFAR-10 | | | | |
|---|---|---|---|---|---|---|---|---|---|---|---|
| | $\bar{C}$ | 0.5 | 0.6 | 0.7 | 0.8 | 0.9 | 0.5 | 0.6 | 0.7 | 0.8 | 0.9 |
| **Gaussian Attack** | H+Clean data | **53.00** | **52.90** | **49.11** | **45.95** | **42.39** | **67.05** | **68.29** | **62.20** | 53.38 | **52.45** |
| | FLTurst | 32.15 | 31.33 | 30.51 | 29.52 | 29.64 | 45.67 | 46.43 | 46.34 | 47.13 | 46.07 |
| | Zeno++ | 39.22 | 36.14 | 37.38 | 35.36 | 33.13 | 46.09 | 54.83 | 42.58 | **57.42** | 8.76 |
| **Sign-flip Attack** | H+Clean data | **53.16** | **52.22** | **49.15** | **45.07** | **42.65** | **66.94** | **63.95** | **59.93** | **59.75** | **54.64** |
| | FLTurst | 22.89 | 22.51 | 23.04 | 25.14 | 25.48 | 40.60 | 34.76 | 34.89 | 31.48 | 39.93 |
| | Zeno++ | 35.58 | 34.30 | 35.25 | 32.82 | 32.84 | 37.36 | 56.46 | 54.50 | 56.65 | 8.76 |
| **LIE Attack** | H+Clean data | **53.63** | **52.24** | **49.44** | **45.67** | **42.61** | **67.01** | **68.05** | **67.39** | **65.60** | **55.75** |
| | FLTurst | 31.45 | 30.92 | 29.81 | 29.92 | 29.59 | 46.21 | 46.30 | 46.32 | 45.98 | 45.43 |
| | Zeno++ | 34.59 | 35.10 | 37.13 | 36.18 | 36.40 | 45.40 | 49.45 | 57.57 | 41.15 | 8.76 |
| **FoE Attack** | H+Clean data | **53.41** | **52.43** | **50.27** | **46.67** | **41.21** | **66.45** | **67.77** | **63.85** | **68.26** | **50.19** |
| | FLTurst | 22.78 | 22.63 | 22.66 | 24.66 | 26.17 | 26.75 | 31.12 | 54.60 | 34.18 | 36.41 |
| | Zeno++ | 34.83 | 32.65 | 35.29 | 35.12 | 14.01 | 56.72 | 57.88 | 32.07 | 48.29 | 8.76 |

achieves SOTA performance across the three benchmarks. Figure 1 illustrates the performance of H+ with clean data (for $\beta = 0.6$ and $\beta = 0.2$) across four attack types on Tiny-ImageNet and CIFAR-100 dataset. More detailed results are provided in Appendix D.

**Method without clean data:** Under **Gaussian attacks**, H+ improves the robustness of Median and Krum in most scenarios. Table 1 shows that H+Krum adapts better to Tiny-ImageNet, with 0.14% – 0.22% higher accuracy than H+Median when $\beta = 0.6$, while H+Median exhibits stronger robustness on CIFAR-100 and CIFAR-10. Notably, H+Median and H+Krum significantly boost the original Median and Krum on Tiny-ImageNet and CIFAR-100, respectively, with accuracy gains of at least 5.97%. At $\beta = 0.2$, H+Median and H+Krum perform comparably, both improving accuracy by at least 4.64% over their base methods. For **Sign-flip attacks**, H+ consistently enhances GM, MCA, and CClip across datasets and data

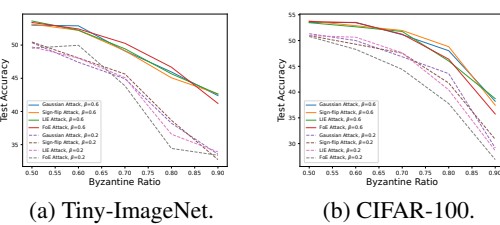

(a) Tiny-ImageNet.  (b) CIFAR-100.

Figure 1: The maximum test accuracy (%) for H+Clean data over five Byzantine ratios on Tiny-ImageNet and CIFAR-100 datasets.

heterogeneity levels. From Table 1, H+CClip outperforms H+GM and H+MCA in most cases (exceptions include $\bar{C} = 0.4$, $\beta = 0.6$ on CIFAR-10 and $\bar{C} = 0.2$, $\beta = 0.2$ on Tiny-ImageNet), demonstrating greater stability against Sign-flip attacks. This suggests the CClip-generated reference vectors better assist H+ in filtering honest vectors. Compared to the original methods, H+ improves accuracy by at least 11.54% for GM, MCA, and CClip under both concentration parameter settings. Under **LIE attacks**, H+Median outperforms H+CClip in most scenarios, particularly at $\beta = 0.2$, indicating stronger adaptability to data heterogeneity. Table 1 confirms significant gains: H+Median improves accuracy by at least 1.29% (at $\beta = 0.6$) and 1.22% (at $\beta = 0.2$) over Median, while H+CClip achieves gains of at least 8.77% (at $\beta = 0.6$) and 11.79% (at $\beta = 0.2$) over CClip. Finally, for **FoE attacks**, H+Krum outperforms H+GM and H+MCA across all three datasets and concentration parameter settings (Table 1), with only a marginal 0.06% accuracy deficit to H+MCA on Tiny-ImageNet at $\beta = 0.2$ and gains of at least 0.18% in all other cases. H+ consistently enhances the original methods: H+Krum improves Krum by at least 31.69%, H+GM improves GM by 11.46%, and H+MCA improves MCA by 7.73%, validating H+'s ability to strengthen existing robust aggregation methods.

In summary, while existing robust methods without clean data often struggle against certain Byzantine attack types or high Byzantine ratios, our H+ method consistently outperforms them, effectively enhancing robustness under these challenging conditions.

**Methods with clean data:** For **Gaussian attacks**, H+ with clean data achieves SOTA accuracy across all five Byzantine ratio settings on Tiny-ImageNet dataset, improving accuracy by at least

9.26% over baselines (Table 2). On CIFAR-10 dataset, it outperforms in four settings, particularly when $\bar{C} = 0.9$. Under **Sign-flip attacks**, H+ with clean data delivers SOTA performance on both two datasets, with accuracy gains of at least 3.1% over baselines. Notably, it excels under high Byzantine ratios (Table 2). In **LIE attacks**, H+ with clean data achieves SOTA accuracy on both Tiny-ImageNet and CIFAR-10, improving test accuracy by at least 6.21% over baselines across all five Byzantine ratios. For **FoE attacks**, H+ with clean data outperforms the baselines by at least 9.25% in accuracy, confirming its SOTA performance.

In summary, as shown in Figure 1 and Table 2, H+ with clean data remains robust across all Byzantine attack types and ratios, while better handling data heterogeneity on simpler datasets. It achieves SOTA performance on Tiny-ImageNet and outperforms baselines on CIFAR-10, especially under high Byzantine ratios.

Table 3: The maximum test accuracy (%) for the H+ method without clean data on $\beta = 0.6$ and Tiny-ImageNet dataset. The best results are in **bold**.

| | $\bar{C}$ | Our Attack | Sign-flip Attack | LIE Attack |
|---|---|---|---|---|
| H+GM | 0.2 | **54.31** | 54.30 | 53.90 |
| | 0.4 | **53.80** | 53.05 | 53.56 |
| H+MCA | 0.2 | **54.23** | 54.20 | 53.75 |
| | 0.4 | **54.14** | 53.39 | 53.66 |
| H+CClip | 0.2 | 54.15 | **54.42** | 54.28 |
| | 0.4 | **53.78** | 53.39 | 53.63 |

Table 4: The maximum test accuracy (%) for the H+ method with clean data on $\bar{C} = 0.6$ and Tiny-ImageNet dataset for three setups of $N$. The best results are in **bold**.

| | $\beta$ | $1.1*M - B$ | $M - B$ | $0.9*M - B$ |
|---|---|---|---|---|
| Gaussian Attack | 0.6 | 52.16 | **52.90** | 51.02 |
| | 0.2 | **49.76** | 47.41 | 46.70 |
| Sign-flip Attack | 0.6 | **53.49** | 52.22 | 51.95 |
| | 0.2 | **48.54** | 48.06 | 46.56 |
| LIE Attack | 0.6 | **52.43** | 52.24 | 51.44 |
| | 0.2 | 46.93 | **48.03** | 44.97 |
| FoE Attack | 0.6 | 50.84 | **52.43** | 49.82 |
| | 0.2 | 47.67 | **49.97** | 45.75 |

### 5.3 ABLATION EXPERIMENT

To evaluate the Byzantine robustness of the similarity check function $H$ independently from the penalty term $\max\{\text{norm}_m, \frac{\tau}{\text{norm}_m}\}$ used in the H+ method, we introduce a tailored Byzantine attack, referred to as "our attack". In this setting, malicious updates are crafted such that their magnitudes closely match those of honest updates, thereby rendering the penalty term ineffective in distinguishing malicious vectors. Details of the attack design are provided in the Appendix D. As shown in Table 3, under "our attack" where the penalty term $\max\{\text{norm}_m, \frac{\tau}{\text{norm}_m}\}$ is rendered ineffective and only the similarity check function $H$ remains active, H+GM, H+MCA, and H+CClip still achieve comparable or even superior performance compared to the cases under Sign-flip and LIE attacks, whose test accuracy in Table 1 represents the mainstream robustness level.

To evaluate the sensitivity of the H+ method to hyperparameter $N$, we conduct an ablation study with three $N$ configurations: $1.1M - B$, $M - B$, and $0.9M - B$. These configurations correspond to $N$ being greater than, equal to, or less than the number of honest clients. As shown in Table 4, the H+ method performs better when $N$ is greater than or equal to the number of honest clients than when $N$ is less than this number; each of these two cases ($N \geq$ honest client count) exhibits distinct strengths and weaknesses across different attacks. Notably, all three configurations outperform the baselines reported in Tables 2 and 7. Thus, the range of valid $N$ values is recommended to be relaxed in practical applications.

In summary, Tables 3 and Table 4 demonstrate that the H+ method robustly defends against Byzantine attacks across diverse complex scenarios.

## 6 CONCLUSION

This paper introduces H+, a similarity-aware aggregation method that enhances FL robustness against Byzantine attacks. It improves performance of existing robust algorithms in the absence of clean data and identifies honest clients when clean data is available. Experiments show that H+ outperforms SOTA methods, offering robust performance across various attack types and datasets, while maintaining low computational complexity.

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

## A    LLM USAGE

We leverage Large Language Model (LLM) to polish the textual content of this paper, including refining sentence structures, enhancing linguistic fluency, and ensuring the accuracy and clarity of academic expressions.

## B    DISCUSSIONS ABOUT H+ ON FL WITHOUT CLEAN DATA

For H+ on FL without clean data, its robustness to attacks depends on the base robustness. Specifically, H+ does not extend the robustness **limits** of the base method, but when the base method already has some robustness against certain attacks, stacking H+ can further improve the overall system's performance. For example, the base method fails under attacks, such as high Byzantine client ratios, H+ provides no benefit. On the other hand, when the base method does not diverge but has bad performance on some specific attacks, H+ can substantially mitigate this weakness, as shown in Section 5.

## C    ALGORITHM WORKFLOW

---

**Algorithm 1** H+ on FL without clean data

---

1: **Input:** Initial global model parameter $w^0$, clients set $\mathcal{M}$, and the number of iteration $T$.
2: **Output:** Updated global model parameter $w^T$.
3: % % **Initialization**
4: Every client $m$ establishes its own set $\mathcal{S}_m$ for $m \in \mathcal{M} \setminus \mathcal{B}$.
5: **for** $t = 0, 1, 2, \cdots, T - 1$ **do**
6:     **for** every client $m \in \mathcal{M} \setminus \mathcal{B}$ in parallel **do**
7:         Receive the global model $w^t$. Select a subdataset $\xi_m^t$ from $\mathcal{S}_m$ to train local model and evaluate the local training gradient $\nabla F_m(w^t, \xi_m^t)$. Set $g_m^t = \nabla F_m(w^t, \xi_m^t)$ and upload $g_m^t$ to the central server.
8:     **end for**
9:     **for** every client $m \in \mathcal{B}$ in parallel **do**
10:        Receive the global model $w^t$. Generate an arbitrary vector or malicious vector $g_m^t$ based on $w^t$, dataset $\mathcal{S}_m$ and other clients. Upload this vector $g_m^t$ to the central server.
11:     **end for**
12:     Receiver all uploaded vectors $g_m^t, m \in \mathcal{M}$. Utilize robust aggregation methods, weight coefficients, and uploaded vectors to calculate $g^t$ by (12).
13:     **for** $k = 1, 2, \cdots, K$ **do**
14:        Randomly select $r$-dimensional segments from the $g^t$ and $g_m^t$. Utilize similarity check function $H$ to calculate the anomaly score by (13), and select $N$ uploaded vectors with highest scores to form set $\mathcal{I}_k^t$.
15:     **end for**
16:     Take the intersection of these $K$ sets as $\mathcal{I}^t$, and update the global model parameter by (15).
17:     Broadcast the model parameter $w^{t+1}$ to all clients.
18: **end for**
19: Output the model parameter $w^T$.

---

## D    EXPERIMENTAL SETUPS AND RESULTS IN DETAIL

To carry out experiments, we set up a machine learning environment in PyTorch 2.3.1 on Ubuntu 20.04, powered by two 3090 GPUs and two Intel Xeon Gold 6226R CPUs. Firstly, we describe the datasets as below:

**Datasets:**

- **Tiny-ImageNet:** The Tiny-ImageNet dataset consists of a training set, a validation set, and a test set. The training set includes 100,000 samples, while both the validation set and the test set contain 10,000 samples each. Each sample is a 64 × 64 pixel color image.

---

**Algorithm 2** H+ on FL with clean data

---

1: **Input:** Initial global model parameter $w^0$, clients set $\mathcal{M}$, and the number of iteration $T$.
2: **Output:** Updated global model parameter $w^T$.
3: % % **Initialization**
4: Every client $m$ establishes its own set $\mathcal{S}_m$ for $m \in \mathcal{M} \setminus \mathcal{B}$ and the central server establishes its own set $\mathcal{S}_0$ if it has clean data.
5: **for** $t = 0, 1, 2, \cdots, T-1$ **do**
6:     **for** every client $m \in \mathcal{M} \setminus \mathcal{B}$ in parallel **do**
7:         Receive the global model $w^t$. Select a subdataset $\xi_m^t$ from $\mathcal{S}_m$ to train local model and evaluate the local training gradient $\nabla F_m(w^t, \xi_m^t)$. Set $g_m^t = \nabla F_m(w^t, \xi_m^t)$ and upload $g_m^t$ to the central server.
8:     **end for**
9:     **for** every client $m \in \mathcal{B}$ in parallel **do**
10:        Receive the global model $w^t$. Generate an arbitrary vector or malicious vector $g_m^t$ based on $w^t$, dataset $\mathcal{S}_m$ and other clients. Upload this vector $g_m^t$ to the central server.
11:     **end for**
12:     Receiver all uploaded vectors $g_m^t, m \in \mathcal{M}$. The central server calculates $g^t$ by (17).
13:     **for** $k = 1, 2, \cdots, K$ **do**
14:        Randomly select $r$-dimensional segments from the $g^t$ and $g_m^t$. Utilize similarity check function $H$ to calculate the anomaly score by (13), and select $N$ uploaded vectors with highest scores to form set $\mathcal{I}_k^t$.
15:     **end for**
16:     Take the intersection of these $K$ sets as $\mathcal{I}^t$, and update the global model parameter by (18) or (19).
17:     Broadcast the model parameter $w^{t+1}$ to all clients.
18: **end for**
19: Output the model parameter $w^T$.

---

- **CIFAR-100:** The CIFAR-100 dataset comprises a training set and a test set. The training set contains 50,000 samples, and the test set contains 10,000 samples, with each sample being a $32 \times 32$ pixel color image. It includes 100 fine-grained classes grouped into 20 broader superclasses, enabling more complex image classification tasks.

- **CIAFR-10:** The CIFAR10 dataset includes a training set and a test set. The training set contains 50,000 samples, and the test set contains 10,000 samples, each of which is a $32 \times 32$ pixel color image.

We split the above three datasets into $M$ non-IID training sets, which is realized by letting the label of data samples to conform to Dirichlet distribution. The extent of non-IID can be adjusted by tuning the concentration parameter $\beta$ of Dirichlet distribution.

**Models:** We adopt MobileNetV3 Howard et al. (2019), VGG16 Simonyan & Zisserman (2014), and ResNet18 He et al. (2016) models, respectively. The introduction of these three models is as follows:

- **MobileNetV3:** MobileNetV3 is a lightweight convolutional neural network (CNN) meticulously optimized for mobile and embedded devices. It integrates depthwise separable convolutions with Neural Architecture Search (NAS) to enable efficient feature extraction and classification under strict computational constraints, with its detailed architectural design documented in Howard et al. (2019). For specific dataset adaptability, we conducted fine-tuning to optimize its performance on the TinyImageNet dataset, ensuring robust feature learning across its 200-class image corpus.

- **VGG16:** The VGG16 model represents a seminal 16-layer convolutional neural network architecture comprising 13 convolutional layers and 3 fully-connected (FC) layers. Each convolutional stage utilizes cascaded 3×3 kernels with stride 1 and ReLU activation, interspersed with 2×2 max-pooling operations that halve spatial resolution while preserving depth. The fully-connected hierarchy consists of two 4,096-unit hidden layers (FC1-2) followed by a 1,000-class output layer (FC3), totaling 138M trainable parameters Simonyan

& Zisserman (2014). For CIFAR-100 dataset adaptation, we implemented fine-tuning to adapt to this dataset.

- **ResNet18:** ResNet18 is a deep convolutional neural network (CNN) featuring 18 weighted layers, distinguished by its innovative residual blocks that alleviate the vanishing gradient problem in deep networks. These blocks enable efficient training of deeper architectures by introducing skip connections, which facilitate the propagation of gradients through the network, as detailed in He et al. (2016). For CIFAR-10 dataset adaptability, we performed fine-tuning to optimize its performance on target datasets, ensuring robust feature learning across diverse image categories.

**Hyperparameters:** We set $M = 50$ and fix the batch size at 32 across all experiments. For numerically computing the GM and MCA, the error tolerance is defined as $\epsilon = 1 \times 10^{-5}$. The concentration parameter $\beta$ takes values 0.6, and 0.2. In all experiments involving the H+ method, we set $K = 3$, $r = 50$, and $N = M - B$ for all experiments. The number of iterations is configured as 100 for these three datasets.

Regarding $\eta^t$, $\rho$, and $\tau$:

- On Tiny-ImageNet dataset, $\eta^t = \frac{0.01}{0.006t+1}$, $\rho = 10$, and $\tau = 0.1$.

- On CIFAR-100 dataset, $\eta^t = \frac{0.004}{0.006t+1}$, $\rho = 10$, and $\tau = 0.1$.

- On CIFAR-10 dataset, $\eta^t = \frac{0.001}{0.006t+1}$, $\rho = 0.1$, and $\tau = 100$.

**Byzantine Attacks:** The ratio of Byzantine attacks, $\bar{C}$, is set to 0.2, 0.4, 0.5, 0.6, 0.7, 0.8 and 0.9. And we select five types of Byzantine attacks, which are introduced as follows,

- **Gaussian attack:** All Byzantine attacks are selected as the Gaussian attack, which obeys $\mathcal{N}(0, 90)$.

- **Sign-flip attack:** All Byzantine clients upload $-3 \cdot \sum_{m \in \mathcal{M} \setminus \mathcal{B}} g_m^t$ or $-3 \cdot \sum_{m \in \mathcal{M}' \setminus \mathcal{B}} g_m^t$ to the central server on iteration number $t$.

- **LIE attack Baruch et al. (2019):** LIE attack adds small amounts of noise to each dimension of the benign gradients. The noise is controlled by a coefficient $c$, which enables the attack to evade detection by robust aggregation methods while negatively impacting the global model. Specifically, the attacker calculates the mean $a$ and standard deviation $\nu$ of the parameters submitted by honest users, calculates the coefficient $c$ based on the total number of honest and malicious clients, and finally computes the malicious update as $a + c\nu$. We set $c$ to 0.7.

- **FoE attack Xie et al. (2020a):** The FoE attack enables Byzantine clients to upload $\frac{q}{M-B} \sum_{\mathcal{M} \setminus \mathcal{B}} g_m^t$ or $\frac{q}{M-B} \sum_{\mathcal{M}' \setminus \mathcal{B}} g_m^t$ to disrupt the FL training process. The coefficient $q$ is configured differently based on the specific attack and algorithm. We set $q = -3 * (M - B)$ for MCA method and $q = -0.1$ for other methods.

- **Our attack:** To ensure attack vectors are close to honest clients' vectors while effectively influencing the FL process, all Byzantine clients upload either $-\frac{1}{M-B} \cdot \sum_{m \in \mathcal{M} \setminus \mathcal{B}} g_m^t$ or $-\frac{1}{M-B+1} \cdot \sum_{m \in \mathcal{M}' \setminus \mathcal{B}} g_m^t$ to the central server at iteration $t$.

**Baselines:** The performance of eight methods (Our method H+, Median, Krum Blanchard et al. (2017), GM, MCA Luan et al. (2024), CClip Karimireddy et al. (2021), FLTrust (Cao et al., 2021), and Zeno++ Xie et al. (2020b)) is compared. Among these, Median, Krum, GM, MCA, and CClip utilize coordinate-wise median, Krum, geometric median, maximum correntropy aggregation, and centered clipping, respectively, to update the global model parameters over the uploaded vectors on FL without clean data. FLTrust and Zeno++ utilizes the clean data on the central server. Note that Cao & Lai (2019) and ByGARS are excluded from comparison due to the lack of open-source code and their relative obsolescence. Among Zeno, Zeno+, and Zeno++, only Zeno++ is evaluated as it is the latest improved version. Our H+ method is evaluated under both frameworks with and without clean data, denoted as H+(X), where X specifies the algorithm to generate the reference vector.

**Metric:** A higher test accuracy indicates better performance and robustness of the robust methods.

**More detailed results:** We show the detailed results about H+ method on different cases in Table 5, Table 6, Table 7 and Table 8.

Table 5: The maximum test accuracy (%) for the H+ method and baselines without clean data on Tiny-ImageNet dataset with $\beta = 0.6$. The best results are in **bold**, and improvements brought by H+ over the original robust methods are underlined.

| Attack Name | Gaussian Attack | | Sign-flip Attack | | LIE Attack | | FoE Attack | |
| --- | --- | --- | --- | --- | --- | --- | --- | --- |
| $\bar{C}$ | 0.2 | 0.4 | 0.2 | 0.4 | 0.2 | 0.4 | 0.2 | 0.4 |
| **H+Median** | 54.67 | 53.23 | 54.39 | 53.17 | 54.61 | 53.95 | 54.31 | 53.34 |
| **Median** | 47.16 | 46.36 | 23.44 | 8.36 | 46.71 | 46.76 | 37.85 | 3.53 |
| **H+Krum** | 54.81 | 53.45 | 54.25 | **53.72** | 54.53 | 53.74 | **54.65** | **54.56** |
| **Krum** | 32.16 | 32.20 | 32.31 | 35.62 | 32.28 | 31.96 | 0.33 | 0.34 |
| **H+GM** | 54.22 | 53.77 | 54.30 | 53.05 | 54.90 | **54.56** | 54.04 | 53.77 |
| **GM** | **54.84** | 54.11 | 42.76 | 0.33 | 55.08 | 53.79 | 42.58 | 0.34 |
| **H+MCA** | 54.83 | **54.65** | 54.20 | 53.39 | 54.75 | 53.98 | 53.91 | 54.02 |
| **MCA** | 54.81 | 54.28 | 0.50 | 0.50 | **55.10** | 53.79 | 0.50 | 0.50 |
| **H+CClip** | 54.28 | 53.76 | **54.42** | 53.39 | 54.28 | 53.63 | 54.91 | 53.70 |
| **CClip** | 45.77 | 40.95 | 36.16 | 0.43 | 45.51 | 40.96 | 34.94 | 0.44 |

Table 6: The maximum test accuracy (%) for the H+ method and baselines without clean data on Tiny-ImageNet dataset with $\beta = 0.2$. The best results are in **bold**, and improvements brought by H+ over the original robust methods are underlined.

| Attack Name | Gaussian Attack | | Sign-flip Attack | | LIE Attack | | FoE Attack | |
| --- | --- | --- | --- | --- | --- | --- | --- | --- |
| $\bar{C}$ | 0.2 | 0.4 | 0.2 | 0.4 | 0.2 | 0.4 | 0.2 | 0.4 |
| **H+Median** | 53.03 | 51.40 | 53.45 | 50.57 | 53.86 | 52.45 | **54.05** | **52.52** |
| **Median** | 21.60 | 23.41 | 0.75 | 16.52 | 22.75 | 22.21 | 14.95 | 3.02 |
| **H+Krum** | **54.16** | 51.37 | **54.01** | 51.06 | **54.17** | **51.64** | 53.37 | 51.48 |
| **Krum** | 26.10 | 25.85 | 26.37 | 25.93 | 25.58 | 26.21 | 0.35 | 0.36 |
| **H+GM** | 53.60 | 52.18 | 52.65 | 51.31 | 53.78 | 50.91 | 53.07 | 49.48 |
| **GM** | 53.59 | 52.76 | 29.64 | 0.06 | 53.57 | 51.42 | 29.78 | 0.35 |
| **H+MCA** | 54.04 | **53.71** | 52.76 | 51.34 | 53.02 | 50.61 | 53.43 | 49.79 |
| **MCA** | 53.46 | 52.89 | 0.51 | 0.50 | 53.70 | 51.45 | 0.51 | 0.50 |
| **H+CClip** | 53.61 | 51.74 | 52.58 | **52.54** | 53.71 | 51.81 | 53.35 | 51.94 |
| **CClip** | 39.37 | 36.56 | 13.92 | 0.41 | 41.98 | 31.79 | 14.15 | 0.43 |

Table 7: The maximum test accuracy (%) for the H+ method and Zeno++ with clean data on $\beta = 0.2$. The best results are in **bold**.

| Attack Name | Datasets | TinyImageNet | | | | | CIFAR10 | | | | |
|---|---|---|---|---|---|---|---|---|---|---|---|
| | $\bar{C}$ | 0.5 | 0.6 | 0.7 | 0.8 | 0.9 | 0.5 | 0.6 | 0.7 | 0.8 | 0.9 |
| **Gaussian Attack** | H+Clean data | **50.36** | **47.41** | **44.93** | **38.23** | **33.56** | **63.79** | **56.94** | **59.16** | **52.63** | 43.09 |
| | FLTrust | 18.91 | 18.95 | 18.96 | 19.27 | 19.26 | 41.54 | 42.37 | 42.38 | 42.20 | **43.21** |
| | Zeno++ | 10.77 | 7.32 | 6.90 | 8.31 | 0.48 | 50.85 | 50.20 | 45.42 | 43.62 | 8.76 |
| **Sign-flip Attack** | H+Clean data | **50.46** | **48.06** | **45.64** | **38.67** | **32.75** | **68.25** | **56.93** | **56.86** | **62.43** | **44.79** |
| | FLTrust | 8.88 | 9.27 | 9.91 | 11.67 | 14.91 | 31.88 | 31.96 | 33.58 | 38.38 | 38.89 |
| | Zeno++ | 7.66 | 8.24 | 9.77 | 5.78 | 1.43 | 53.39 | 51.11 | 45.68 | 42.73 | 8.76 |
| **LIE Attack** | H+Clean data | **49.72** | **48.03** | **45.09** | **36.57** | **33.91** | **62.89** | **61.09** | **64.67** | **55.96** | **46.79** |
| | FLTrust | 19.04 | 18.90 | 18.71 | 19.08 | 19.19 | 42.14 | 43.12 | 41.75 | 43.40 | 43.53 |
| | Zeno++ | 16.21 | 10.47 | 8.68 | 0.48 | 1.38 | 50.32 | 45.54 | 48.25 | 43.64 | 8.76 |
| **FoE Attack** | H+Clean data | **49.55** | **49.97** | **43.86** | **34.43** | **33.40** | **61.92** | **72.06** | **68.62** | **37.21** | 12.89 |
| | FLTrust | 9.18 | 9.07 | 9.96 | 11.72 | 15.45 | 33.75 | 31.75 | 34.95 | 36.30 | 40.36 |
| | Zeno++ | 10.62 | 5.76 | 6.73 | 6.06 | 5.00 | 58.99 | 50.66 | 54.34 | 26.51 | **41.56** |

Table 8: The maximum test accuracy (%) for the H+ method and Zeno++ with clean data on CIFAR-100 dataset. The best results are in **bold**.

| Attack Name | $\beta$ | 0.6 | | | | | 0.2 | | | | |
|---|---|---|---|---|---|---|---|---|---|---|---|
| | $\bar{C}$ | 0.5 | 0.6 | 0.7 | 0.8 | 0.9 | 0.5 | 0.6 | 0.7 | 0.8 | 0.9 |
| **Gaussian Attack** | H+Clean data | **53.50** | **53.53** | **51.14** | **48.03** | 38.22 | **51.33** | **50.00** | **46.85** | **43.58** | **29.17** |
| | FLTrust | 31.62 | 31.65 | 31.00 | 30.82 | 29.66 | 20.70 | 20.76 | 20.60 | 20.43 | 20.46 |
| | Zeno++ | 39.73 | 37.95 | 41.73 | 39.37 | **38.63** | 30.79 | 29.26 | 31.20 | 28.82 | 25.10 |
| **Sign-flip Attack** | H+Clean data | **53.80** | **52.88** | **51.96** | **48.81** | **37.46** | **50.96** | **49.29** | **47.56** | **41.76** | 30.84 |
| | FLTrust | 24.90 | 26.27 | 25.32 | 27.30 | 29.22 | 17.94 | 17.75 | 17.71 | 19.07 | 19.71 |
| | Zeno++ | 37.58 | 37.87 | 34.09 | 38.37 | 37.16 | 30.08 | 24.51 | 27.02 | 29.82 | **31.68** |
| **LIE Attack** | H+Clean data | **53.47** | **52.67** | **51.76** | **46.03** | **38.73** | **50.97** | **50.65** | **47.61** | **40.43** | **28.64** |
| | FLTrust | 31.45 | 31.04 | 31.33 | 30.18 | 29.50 | 20.71 | 20.92 | 20.47 | 20.46 | 20.34 |
| | Zeno++ | 33.73 | 39.70 | 37.73 | 34.77 | 35.88 | 30.52 | 28.08 | 26.40 | 21.19 | 24.02 |
| **FoE Attack** | H+Clean data | **53.71** | **53.47** | **51.23** | **46.39** | 35.75 | **50.79** | **48.29** | **44.41** | **37.77** | **26.90** |
| | FLTrust | 25.46 | 25.54 | 26.41 | 26.07 | 28.18 | 18.17 | 18.26 | 18.53 | 18.84 | 20.23 |
| | Zeno++ | 40.59 | 40.07 | 38.09 | 36.61 | **38.84** | 26.32 | 31.71 | 27.68 | 28.84 | 25.07 |

