# OpenReview forum: "H+: An Efficient Similarity-Aware Aggregation for Byzantine Resilient Federated Learning"
_ICLR.cc/2026/Conference — ICLR 2026 Conference Withdrawn Submission_

### Official Review · Reviewer_frR2 · 2025-10-18

**Soundness:** 2
**Presentation:** 2
**Contribution:** 2
**Rating:** 2
**Confidence:** 4

**Summary:**

This paper proposes H+, a similarity-aware aggregation method aimed at improving the robustness of federated learning (FL) against Byzantine attacks. The approach enhances existing robust aggregation algorithms by modeling client similarity both with and without access to clean reference data. When clean data is available, H+ can further identify and prioritize honest clients, improving resilience to adversarial behavior.

**Strengths:**

1. The proposed method is easy to follow.
2. Extensive experiments were conducted.

**Weaknesses:**

1. The idea of using “repeated slicing” to mitigate the curse of dimensionality is not new and has already been proposed in DnC (NDSS 2021). The paper does not clearly explain how its approach differs conceptually or technically from prior work, which raises concerns about the novelty of this contribution.

2. The paper introduces a new similarity check function H, but does not explain its underlying intuition or provide insight into why it performs better than standard similarity metrics such as cosine similarity or Euclidean distance. Without a clear theoretical or empirical justification, it is difficult to assess the significance of this design choice.

3. All evaluated attacks are from 2020 or earlier, which limits the relevance of the experimental results. Considering the submission aims for ICLR 2026, the absence of more recent and adaptive attack baselines is surprising and weakens the claimed effectiveness of the proposed method.

4. The paper provides no theoretical discussion or analysis regarding the effectiveness of the similarity function H or the potential impact of the proposed defense on model convergence. Such analysis would be critical to support the soundness and stability of the approach.

5. It remains unclear whether the proposed method generalizes to modern model architectures such as Vision Transformers (ViT). Given the growing dominance of ViTs in vision-related FL tasks, this omission limits the scope and practical relevance of the work.

6. As highlighted by Back to the Drawing Board: A Critical Evaluation of Poisoning Attacks on Production Federated Learning (IEEE S&P 2022), the proportion of malicious clients in real-world federated deployments is typically below 1%. Under such conditions, Byzantine attacks pose little practical threat. Thus, assuming more than 50% of clients are malicious is unrealistic and substantially reduces the real-world value of the proposed evaluation.

**Questions:**

Please refer to Weaknesses.

---

### Official Review · Reviewer_pF77 · 2025-10-31

**Soundness:** 2
**Presentation:** 2
**Contribution:** 2
**Rating:** 2
**Confidence:** 5

**Summary:**

The authors present a new similarity-aware aggregation method to enhance the security of federated learning.

**Strengths:**

1. The paper presents a novel method aimed at enhancing the security of federated learning systems.

2. Experimental results validate the proposed method’s effectiveness.

**Weaknesses:**

1. If the base method fails, the proposed approach also fails.

2. The concept of enhancing the robustness of existing defense methods has already been explored in previous studies.

3. The attacks evaluated in the paper are relatively weak and do not represent strong or adaptive adversaries.

4. The paper does not clearly describe the details of clean data used in the experiments.

5. The work lacks formal theoretical guarantees.

**Questions:**

1. In cases where clean data is unavailable, which is common in real-world federated learning since obtaining clean data is often difficult, the proposed H+ method cannot extend the robustness boundary of the underlying aggregation rule. It only enhances performance when the base method (for example, Krum, GM, or Median) already provides some level of resilience. However, when the base method fails under strong attacks, H+ also fails. This greatly limits its effectiveness. For instance, as shown in [a], the Krum aggregation rule is inherently vulnerable to the Krum attack, meaning that Krum provides no robustness under such attack. In this situation, the H+ method will also fail.

2. When clean data is unavailable, H+ can only improve performance if the base method already offers partial robustness. This idea has already been explored in paper [b], which also focuses on improving the robustness of existing aggregation methods such as GM and Median. The authors should clearly explain how their method differs from [b] and include a direct experimental comparison.

3. The attacks considered in this paper are relatively weak. The authors should evaluate their method against state-of-the-art poisoning attacks in federated learning, such as those proposed in [a] and [c].

4. When clean data is available, the paper does not specify its size or characteristics. It remains unclear whether the clean data can be arbitrary or must be sampled from the same distribution as the clients’ training data.

5. The paper only presents empirical results to demonstrate robustness and does not provide any theoretical guarantees or convergence analysis under adversarial conditions.



[a] Local Model Poisoning Attacks to Byzantine-Robust Federated Learning. In USENIX Security Symposium 2020.

[b] Do We Really Need to Design New Byzantine-robust Aggregation Rules. In NDSS 2025.

[c] A Little Is Enough Circumventing Defenses For Distributed Learning. In NeurIPS 2019.

---

### Official Review · Reviewer_8Kvh · 2025-10-31

**Soundness:** 2
**Presentation:** 3
**Contribution:** 2
**Rating:** 2
**Confidence:** 4

**Summary:**

The paper introduces H+, a defense against poisoning attacks in federated learning settings. The method is capable of working in two different scenarios: 1) when the aggregator has its own trusted dataset (or alternatively, there is a number of trusted clients with clean datasets); 2) there is no trusted dataset or a trusted set of clients. The defense relies on a similarity metric that is computed on random subsets of the parameters of the model, allowing to speed up the computation and to reduce the effect of the curse of dimensionality. The experimental evaluation includes different computer vision benchmarks, models, and defenses, both in scenarios with and without clean data. In these settings, H+ shows competitive results and outperforms other defenses in the related work.

**Strengths:**

+ The defense work in scenarios where the aggregator has a clean/trusted dataset available and in scenarios with no trusted data, which is uncommon for defenses against poisoning attacks in federated learning, who typically rely on one or the other assumption.
+ The proposed method is also modular and allows to combine it with other aggregation methods, including robust aggregation with KRUM, MCA, median, etc.
+ The authors also prioritized computational efficiency, reducing the number of parameters used for the computation of the similarity metric.

**Weaknesses:**

+ The main limitation of the proposed defense is that it requires to provide the expected number of benign participants and, throughout most experiments the authors assume that this number is known in advance, which is unrealistic for practical scenarios, where the number of attackers is unknown and, also, can vary throughout the training of the algorithm. The ablation study only explores scenarios with ±10% deviations in the number of benign clients, which barely tests robustness to practical uncertainty.

+ Following the previous point, the paper lacks evaluation in benign conditions, i.e., when there are no attackers. This is important as we cannot assess whether H+ introduces any degradation, noise, or training instability when all clients are honest and we only select a subset of N clients at each training iteration. In this sense, a robust aggregation method should aim to keep the baseline model’s accuracy in the absence of attacks.

+ The similarity metric in equation (10) is not well justified and discussed. Why is this metric appropriate compared to other existing similarity metrics, like, for example, the cosine similarity. Is there any advantage in using a non-symmetric similarity metric like the one in (10), compared to symmetric similarity metrics? I think this point would make the paper sounder.

 + There is a relevant dependence with respect to the different hyperparameters of the method. In this sense, the method uses fixed hyperparameters with minimal justification, so that, generalization across datasets remain unclear, and the ablation studies do not provide a clear view of the sensitivity to these hyperparameters.

+ In the experiments the authors just considered scenarios with 50 clients, limiting the scope of the analysis to, for example, scenarios with more clients, where scalability starts to be a more important aspect to consider.

**Questions:**

+ Based on the comments in the previous sections, it would be interesting to observe how H+ performs when there is no attack and to have a deeper analysis of the effect of N (the number of benign clients) in scenarios where the defender knows little about the potential number of attackers.

+ How do the authors justify the use of (10) as a metric to compute the similarities compared to other metrics?

---

### Official Review · Reviewer_aGBA · 2025-11-01

**Soundness:** 3
**Presentation:** 2
**Contribution:** 3
**Rating:** 6
**Confidence:** 3

**Summary:**

This paper introduces H+, a similarity-aware aggregation method designed for byzantine-resilient federated learning (FL). The method leverages a novel similarity check function, H, which measures the alignment of client-uploaded model parameter segments with a reference vector, intended to distinguish honest clients from byzantine attackers. H+ is adaptable: in systems with clean data, the reference is derived from the clean set; when such data is unavailable, outputs from robust aggregators are used as references. The paper claims improved computational efficiency ($\mathcal{O}(KMr)$) compared to existing similarity-based approaches and empirically demonstrates H+ outperforms or enhances baselines across several attack types, Byzantine ratios, and datasets (Tiny-ImageNet, CIFAR-100, CIFAR-10), with results detailed in extensive tables and visualized in figures.

**Strengths:**

1. Methodological Generality: The H+ framework provides a unified approach to similarity-aware aggregation in federated learning, accommodating both scenarios—with and without access to clean reference data. This flexibility extends its practical relevance beyond the limitations of earlier methods that operate only in clean-data settings.
2. Computational Efficiency: By employing random sampling of low-dimensional vector segments (where 𝑟 ≪ 𝑝) for similarity computation, H+ significantly reduces the computational cost compared to traditional cosine-similarity-based aggregation, making it well-suited for large-scale models.
3. Empirical Robustness: The experimental results demonstrate that H+ consistently enhances test accuracy and resilience against a variety of Byzantine attack types including Gaussian, Sign-flip, LIE, and FoE across a broad range of attack ratios. It often achieves substantial performance gains over existing robust aggregators such as Median, Krum, GM, MCA, CClip, FLTrust, and Zeno++.
4. Ablation and Sensitivity Analysis: The inclusion of comprehensive ablations and sensitivity studies effectively isolates the contributions of key components (e.g., the H function and the 𝑁 hyperparameter), reinforcing the credibility and robustness of the empirical findings.
5. Clarity and Analytical Rigor: The methodology is clearly presented through both descriptive explanations and pseudocode, with accompanying analytical discussions that justify the time complexity and design choices, ensuring transparency and reproducibility.

**Weaknesses:**

1. Some more recent works working on mitigating Byzantine attacks should be surveyed and cited.
Xu, J., Zhang, Z., Hu, R., Achieving Byzantine-Resilient Federated Learning via Layer-Adaptive Sparsified Model Aggregation (2024) ; etc
2. Complexity in Evaluation Setup and Limited Dataset Diversity: The experimental evaluation primarily focuses on image classification tasks using well-established, moderately scaled datasets such as Tiny-ImageNet and CIFAR-100/10, with up to 50 clients. While the results are promising, the study relies on conventional model architectures (MobileNetV3, VGG16, ResNet18) and does not provide evidence of scalability to more complex domains such as NLP, time series, or federated tabular data.

**Questions:**

NA

---

### Note · Authors · 2025-11-21

I have read and agree with the venue's withdrawal policy on behalf of myself and my co-authors.